# Additive effect of diabetes mellitus on the prevalence and prognosis of sarcopenic obesity: Implications for all-cause mortality

Ting-Ju Kuo [ID][1]☯, Shih-Wei Huang [ID][2,3]☯, Hui-Wen Lin [ID][4]*

1 Department of Physical Medicine and Rehabilitation, Shuang Ho Hospital, Taipei Medical University, Taipei, Taiwan, 2 Department of Physical Medicine and Rehabilitation, School of Medicine, College of Medicine, Taipei Medical University, Taipei, Taiwan, 3 Department of Physical Medicine and Rehabilitation, Wan Fang Hospital, Taipei Medical University, Taipei, Taiwan, 4 Department of Mathematics, Soochow University, Taipei, Taiwan

☯ These authors contributed equally to this work.
* hwlin@scu.edu.tw

## Abstract

Diabetes mellitus (DM) and sarcopenic obesity are common conditions associated with increased morbidity and mortality. DM, characterized by chronic hyperglycemia, is a recognized risk factor for cardiovascular disease and premature death. Sarcopenic obesity, characterized by reduced muscle mass and increased adiposity, contributes to physical frailty and metabolic dysfunction. This study investigated the effect of DM on mortality rates and causes of death among individuals with high adiposity and low muscle mass (HA-LM) by using data from the National Health and Nutrition Examination Survey (NHANES) linked to mortality records from 2011 to 2018. A total of 2366 patients with HA-LM patients were analyzed, including 194 (8.199%) with DM and 2172 (91.80%) without DM. During the study period, the mortality rate was 1.19% in the HA-LM without DM group and 5.15% in the HA-LM with DM group. Kaplan-Meier survival analysis demonstrated a significantly higher mortality rate in the HA-LM patients with DM group, supported by both crude (hazard ratio [HR]: 4.34, 95% confidence interval [CI]: 2.09–9.00, p < 0.001) and adjusted (HR: 2.88, 95% CI: 1.23–6.73, p < 0.01) models. Cause-specific analysis revealed that heart disease (40%) was the leading cause of mortality in the HA-LM with DM group, followed by other residual causes (30%). By contrast, other residual causes were predominant among those without DM (34.62%), followed by malignant neoplasms (19.23%). These findings underscore the synergistic effects of DM and sarcopenic obesity on the risk of mortality and emphasize the need for targeted interventions aimed at managing diabetes and preserving muscle mass. The study findings may inform interventions aimed at improving health outcomes and reducing mortality in this high-risk population.

**Data availability statement:** All data used in this study are publicly available from the National Health and Nutrition Examination Survey (NHANES) and are linked to the National Death Index (NDI) mortality records. NHANES datasets can be accessed through the U.S. Centers for Disease Control and Prevention (CDC) at: https://www.cdc.gov/nchs/nhanes/. The linked mortality data are available at: https://www.cdc.gov/nchs/data-linkage/mortality.htm.

**Funding:** The author(s) received no specific funding for this work.

**Competing interests:** The authors have declared that no competing interests exist.

## Introduction

Diabetes mellitus (DM) [1] and sarcopenic obesity [2,3] are major health concerns associated with increased morbidity and mortality. DM, characterized by chronic hyperglycemia, is a well-established risk factor for cardiovascular disease [4], renal complications [5], and premature mortality [6,7]. Sarcopenic obesity, characterized by the concurrent loss of muscle mass loss and the accumulation of adipose tissue, contributes to physical frailty and functional decline. This condition also increases patients' susceptibility to various metabolic disorders, further increasing the risk of all-cause mortality [8–10].

The coexistence of DM and sarcopenic obesity may produce a synergistic effect that worsens health outcomes, particularly mortality. In a longitudinal study, Song et al. [11] examined the combined effect of DM and sarcopenia, and reported that their simultaneous presence substantially increased the risks of both all-cause and cardio-vascular mortality. These findings underscores the importance of specifically examining the effect of DM on the prognosis of sarcopenic obesity.

To address the aforementioned gap, we explored the additive effect of DM on mortality in individuals with sarcopenic obesity by using nationally representative data. We also examined cause-specific mortality data to precisely define the risk profile of this population. By elucidating these relationships, our goal was to inform future prevention and intervention strategies targeting individuals affected by the dual burden of DM and sarcopenic obesity.

## Materials and methods

### Study design and population

Secondary data were obtained from the National Health and Nutrition Examination Survey (NHANES), which was administered by the National Center for Health Statistics (NCHS) under the Centers for Disease Control and Prevention (CDC) from 2011 to 2018. The NHANES integrates interviews and physical examinations to evaluate the health and nutritional profiles of individuals across various age groups. The NCHS provides researchers with access to NHANES data for research purposes.

To ensure patient confidentiality and data integrity, NHANES data undergo rigorous vetting and validation before being released to researchers. This dataset serves as a valuable resource for conducting public health research and policy development, offering longitudinal insights into health trends and disparities across various demographic groups in the United States.

This study employed cross-sectional data from the NHANES, including results of whole-body dual-energy X-ray absorptiometry (DXA) scans conducted during the specified period. DXA datasets are publicly available on the Centers for Disease Control and Prevention website (http://www.cdc.gov/nchs/about/major/nhanes/dxx/dxa.htm). The scans were performed using a QDR 4500-A fan beam densitometer (default configuration, software version 12.1; Hologic), ensuring minimal radiation exposure (less than 10 microsieverts) for participant safety. Key variables analyzed included the fat mass index (FMI) and appendicular skeletal muscle mass index (ASMI), both derived from DXA measurements of fat mass and lean soft tissue, respectively.

Although DXA scans are available for NHANES participants aged 8–59 years, we restricted our analysis to adults aged 18–59 years to focus on the adult population and their conditions. Participants were excluded if they had missing data on diabetes status, were ineligible for mortality linkage, or had incomplete covariate information—standard exclusions in secondary data analysis to ensure analytical validity. To evaluate the potential for selection bias, we compared key characteristics (e.g., age, sex, and race or ethnicity) in individuals included in the final analytic sample and those excluded. These groups exhibited similar distributions across these characteristics, indicating that the exclusions may not have introduced meaningful bias or compromised the generalizability of our findings.

Body composition phenotypes, reflecting varying levels of muscularity and adiposity, were categorized using the method proposed by Baumgartner [12]. The revised classification [13] stratifies the population by body mass index(BMI) and sex, establishing specific cutoffs to define four distinct body composition phenotypes: low adiposity with high muscle mass (LA-HM), high adiposity with high muscle mass (HA-HM), low adiposity with low muscle mass (LA-LM), and high adiposity with low muscle mass (HA-LM). The cutoff points which are determined using deciles are as follows: LA-HM (ASMI: 50–100; FMI: 0–49.99), HA-HM (ASMI: 50–100; FMI: 50–100), LA-LM (ASMI: 0–49.99; FMI: 0–49.99), and HA-LM (ASMI: 0–49.99; FMI: 50–100).

## Data collection

The NHANES collects comprehensive health-related information from a diverse sample of the US population sample through interviews, physical examinations, and laboratory tests. The demographic parameters measured included BMI, triceps and subscapular skinfold thickness, ASMI, FMI, age, sex, hypertension and race. The biochemical profile included albumin, alkaline phosphatase, blood urea nitrogen (BUN), creatinine, triglycerides, glycohemoglobin, and total cholesterol. The systemic immune-inflammation index (SII) [14] was calculated using the following formula: P (absolute platelet count) × N (absolute neutrophil count)/L (absolute lymphocyte count).

**Study outcome.** Mortality data for the NHANES participants were obtained from publicly available linked mortality files which provide follow-up mortality information from the date of survey participation. The outcomes of interest included all-cause mortality and deaths from specific causes, namely cardiovascular disease, malignant neoplasms, chronic lower respiratory diseases, accidents (unintentional injuries), cerebrovascular diseases, Alzheimer's disease, DM, influenza, pneumonia, nephritis, nephrotic syndrome, nephrosis, and others causes.

## Statistical analysis

Baseline characteristics were compared in participants in the HA-LM group with and without DM. Continuous variables were compared using weighted *t* tests or Wilcoxon's rank-sum tests, as appropriate, and are reported as mean±standard error. Categorical variables were compared using chi-square tests and are presented as unweighted counts (with weighted percentages). A two-sided *p* value of <0.05 was considered statistically significant.

After a survival analysis was conducted, Kaplan–Meier survival curves were constructed to estimate cumulative survival over time for the two groups (HA-LM with DM vs. HA-LM without DM), with a log-rank test used to compare survival distributions. Time zero was set as the NHANES examination date, and time was measured in months since enrollment.

To quantify the association between DM and mortality, Cox proportional-hazards models were employed. Initially, a crude (unadjusted) hazard ratio (HR) was calculated, followed by a multivariable-adjusted HR to control for potential confounders. Covariates were then selected on the basis of subject-matter knowledge and bivariate analyses and included the following: age, sex, race or ethnicity, BMI, hypertension status, SII, albumin, alkaline phosphatase, BUN, creatinine, triglycerides, and total cholesterol. The ASMI and FMI were excluded from the Cox model because all participants had already met the HA-LM phenotype criteria, and their inclusion may introduce multicollinearity with BMI.

The proportional-hazards assumption for the Cox model was tested by examining Schoenfeld residuals and log(−log) survival plots for each covariate. No significant violation of the proportional-hazards assumption was observed.

Specifically, the Schoenfeld residual tests were nonsignificant for all model covariates (with a threshold of $p < 0.05$), and the log(−log) plots demonstrated roughly parallel curves for the DM versus non-DM groups, indicating that the HRs were relatively constant over time.

A cause-specific mortality analysis was conducted using cumulative incidence functions for each major cause of death, with other causes treated as competing risks. However, given the relatively small number of deaths in certain categories, these results are presented descriptively (percentages) rather than through formal competing risk models.

All statistical analyses were conducted using SAS software version 9.4 (SAS Institute, Cary, NC, USA) and STATA version 12.0 (StataCorp, College Station, TX, USA).Results

A total of 4644 individuals were initially identified from the NHANES data set from 2011 to 2018 as meeting the HA-LM criteria. After cross-referencing with DM test data, 2981 individuals remained. Subsequent linkage with mortality follow-up data resulted in the retention of 2412 individuals. After the exclusion of cases with missing values for various covariates, such as age, sex, and the SII, the final sample comprised 2366 individuals.

**Baseline characteristics.** This study included 2366 patients with HA-LM, of whom 194 (8.199%) had DM, and 2172 (91.80%) did not. Table 1 presents the baseline characteristics of both groups. Patients in the HA-LM with DM group were significantly older than those in the HA-LM without DM group ($p < 0.001$). Compared with the HA-LM without DM group, the HA-LM with DM group had significantly lower albumin levels ($p < 0.001$) and significantly higher levels of. alkaline phosphatase, BUN, creatinine, triglycerides, and glycohemoglobin (all $p < 0.001$). Furthermore, BMI was significantly higher in the HA-LM with DM group than in the HA-LM without DM group. However, no significant between-group differences were observed in terms of the SII, subscapular and triceps skinfold thickness, ASMI or FMI ($p < 0.001$). Hypertension was more prevalent in the HA-LM with DM group than in the HA-LM without DM group ($p < 0.001$).

**Impact of sarcopenic obesity with DM on mortality and analysis of cause-specific mortality.** Among patients with HA-LM, the mortality rate was 1.19% in those without DM and 5.15% in those with DM. As shown in Fig 1, Kaplan–Meier survival analysis revealed a considerably higher mortality rate over the follow-up period in the HA-LM with DM group than in the HA-LM without DM group. Patients with DM exhibited a significantly increased risk of mortality, with a HR of 4.34 (95% CI: 2.09–9.00, $p < 0.001$) and with an adjusted HR of 2.88 (95% CI: 1.23–6.73, $p < 0.01$) after adjustment for confounders (Table 2).

Cause-specific mortality data are presented in Fig 2. In the HA-LM with DM group, heart disease was the leading cause of death, accounting for 40% of the cases, followed by other residual causes (30%). By contrast, the primary cause of death in the HA-LM without DM group was other residual causes, accounting for 34.62% of the cases, followed by malignant neoplasms (19.23%).

## Discussion

This study investigated the effect of DM on the prevalence and prognosis of sarcopenic obesity as well as its association with all-cause mortality by using NHANES data linked with mortality records. Our crude and adjusted HRs indicated that patients with HA-LM and DM exhibited a significantly higher risk of mortality than did patients with HA-LM but without DM. The adjusted HRs underscored the independent and substantial effect of DM on mortality, even after accounting for age, sex, and other potential confounding factors.

Cardiovascular disease was identified as the leading cause of mortality among patients with HA-LM and DM, reinforcing the well-established association between DM and cardiovascular complications, including coronary artery disease, myocardial infarction, and heart failure [15]. Among the pathophysiological mechanisms linking DM to cardiovascular disease are chronic hyperglycemia, insulin resistance, and systemic inflammation, all of which contribute to accelerated atherosclerosis and vascular damage [16].

In the present study, we observed that although patients with HA-LM and DM exhibited an increased risk of cardiovascular mortality, they did not exhibit a significantly increased risk of cerebrovascular diseases, such as stroke. This finding

**Table 1. Demographic and clinical characteristics of participants with HA-LM without and with DM group.**

| | HA-LM without DM (N = 2172) | | HA-LM with DM (N = 194) | | |
|---|---|---|---|---|---|
| | Mean | Std. Error of Mean | Mean | Std. Error of Mean | P value |
| Body mass index (kg/m²) | 28.55 | 0.157 | 35.25 | .604 | <0.001 |
| Systemic immune-inflammation index (SII) | 503.41 | 5.936 | 569.82 | 28.632 | 0.002 |
| Triceps Skinfold thickness(mm) | 23.61 | 0.130 | 23.27 | .424 | 0.455 |
| Subscapular Skinfold thickness (mm) | 21.16 | 0.140 | 21.17 | .457 | 0.985 |
| ASMI | 6.21 | 0.014 | 6.23 | .046 | 0.663 |
| FMI | 10.89 | 0.047 | 10.88 | .151 | 0.990 |
| Age | 37.32 | 0.266 | 47.66 | .678 | <0.001 |
| Albumin (g/dL) | 4.27 | 0.008 | 4.11 | .028 | <0.001 |
| Alkaline phosphatase (U/L) | 69.17 | 0.491 | 76.91 | 1.775 | <0.001 |
| Blood urea nitrogen (mg/dL) | 12.30 | 0.092 | 14.64 | .566 | <0.001 |
| Creatinine (mg/dL) | .84 | 0.005 | .99 | .093 | <0.001 |
| Triglycerides (mg/dL) | 141.67 | 2.839 | 224.08 | 18.752 | <0.001 |
| Glycohemoglobin (%) | 5.45 | 0.013 | 7.83 | .158 | <0.001 |
| Total cholesterol (mg/dL) | 189.09 | 0.875 | 188.32 | 3.699 | 0.805 |
| Sex | | | | | 0.637 |
| Male | 1046 (92.1%) | | 90 (7.9%) | | |
| Female | 1126 (91.5%) | | 104 (8.5%) | | |
| Hypertension | | | | | <0.001 |
| Yes | 441 (80.2%) | | 109 (19.8%) | | |
| No | 1731 (95.3%) | | 85 (4.7%) | | |
| Race | | | | | 0.002 |
| Mexican American | 319 (88.6%) | | 41 (11.4%) | | |
| Other Hispanic | 227 (93.8%) | | 15 (6.2%) | | |
| Non-Hispanic White | 722 (93.0%) | | 54 (7.0%) | | |
| Non-Hispanic Black | 482 (89.8%) | | 55 (10.2%) | | |
| Non-Hispanic Asian | 328 (95.3%) | | 16 (4.7%) | | |
| Other Race | 94 (87.9%) | | 13 (12.1%) | | |

HA-LM: high adiposity and low muscle mass, ASMI: appendicular skeletal muscle mass index, FMI: fat mass index

is inconsistent with the well- established association between DM and an increased risk of ischemic stroke [17], which is typically attributed to microvascular complications [5], thromboembolic events [4], and hypertension [18] associated with diabetes. Overall, the lack of a significant association in our analysis may be attributable to various factors, such as sample size limitations, heterogeneity in diabetes severity, and differences in the classification and ascertainment of stroke-related mortality.

In this study, patients with HA-LM but without DM exhibited a different mortality profile, with the majority of deaths attributable to "all other causes" and malignant neoplasms. This finding indicates that, in the absence of DM, mortality may be driven by different underlying mechanisms, such as aging-related decline, cancer-related complications, and other chronic conditions commonly observed in this population.

Although this study primarily examined mortality outcomes, the associations between DM and markers of sarcopenic obesity were also explored. Patients with HA-LM and DM exhibited higher BMI than did those with HA-LM but without DM. This finding is consistent with the reported association between DM and obesity [19]. However, we did not observe significant differences in the ASMI or FMI between the groups. This finding indicates that although DM influences overall

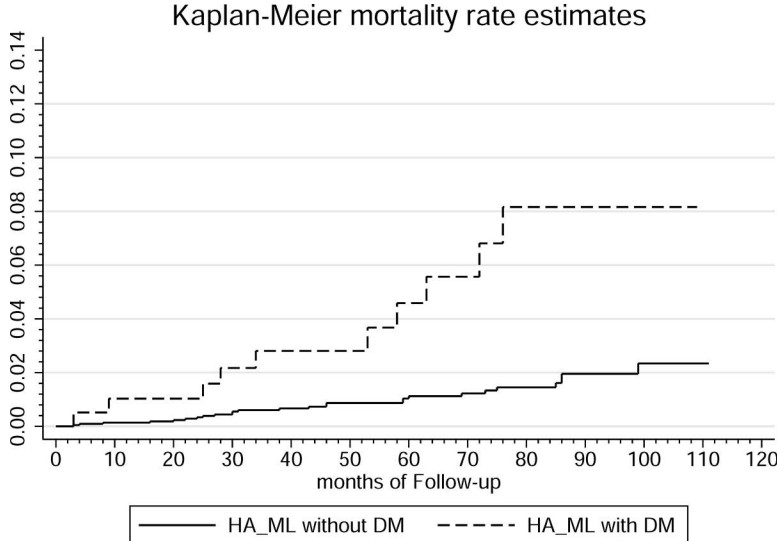

**Fig 1. Kaplan-Meier survival estimates of mortality.**

**Table 2. Crude and adjusted hazard ratios (HRs) and 95% confidence intervals (CIs) for mortality during the person-months of follow-up from NHANES.**

| Presence of mortality | Patients with HA-LM | |
|---|---|---|
| | without DM | with DM |
| Person-months of Follow-up (Mean) | 62.11 | 61.73 |
| Mortality rate | 26/2172 (1.19%) | 10/194 (5.15%) |
| Crude HR (95% CI) | 1.00 | 4.34*** (2.09-9.00) |
| Adjusted HR (95% CI) | 1.00 | 2.88** (1.23-6.73) |

Values are adjusted for demographic characteristics and comorbid conditions through propensity scores, including age, sex, race, body mass index, systemic immune inflammation index, hypertension, albumin (g/dL), alkaline phosphatase (U/L), blood urea nitrogen (mg/dL), creatinine (mg/dL), triglycerides (mg/dL), total Cholesterol (mg/dL).

** p<0.01, *** p<0.001.

changes in body composition, it may not specifically exacerbate the symptoms of sarcopenic obesity beyond the contributions of aging and other concurrent comorbidities.

A previous study reported negative associations between skeletal muscle mass and DM incidence, insulin resistance, and HbA1c levels in healthy adults [20]. This phenomenon indicates the metabolic implications of DM on muscle health, in which insulin resistance and chronic hyperglycemia may contribute to progressive muscle wasting. By contrast, our findings suggest that, in the context of sarcopenic obesity, characterized by concurrent muscle mass loss and excess adiposity, the effect of DM on muscle mass may be mitigated or counterbalanced by other factors, such as age-related sarcopenia and obesity-related metabolic changes.

The concept of the "obesity paradox" [8] complicates the interpretation of the relationship between BMI and body composition.. This paradox suggests that higher BMI in older adults, typically indicative of obesity, may confer protective effects on muscle mass and strength [21]. This paradoxical relationship underscores the limitations of using BMI alone to assess body composition in individuals with DM, as it may not adequately represent the relative contributions of lean muscle mass and fat mass.

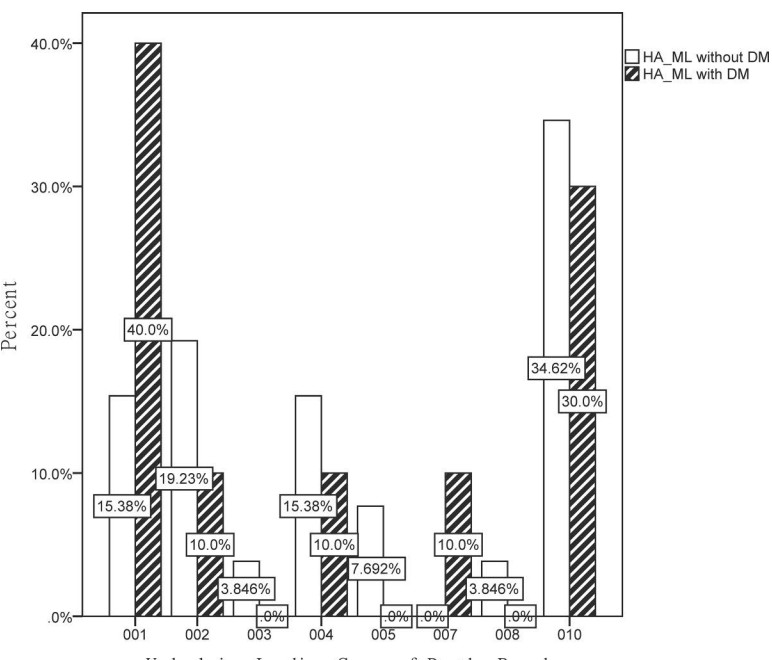

**Fig 2. Causes of mortality among patients with HA-LM with and without DM.** Cause of Death: Recode: 001 Heart disease (054–068). 002 Malignant neoplasms (019–043). 003 Chronic lower respiratory diseases (082–086). 004 Accidents (unintentional injuries) (112–123). 005 Cerebrovascular diseases (070). 006 Alzheimer's disease (052). 007 Diabetes mellitus (046). 008 Influenza and pneumonia (076–078). 009 Nephritis, nephrotic syndrome and nephrosis (097–101). 010 All other causes (residual).

Our findings revealed major between-group differences in metabolic profiles. Specifically, patients with HA-LM and DM exhibited a significantly lower albumin levels but higher BUN and creatinine levels than did patients with HA-LM but without DM. Albumin is essential for maintaining oncotic pressure, transporting hormones and fatty acids, and modulating antioxidant and anti-inflammatory processes [22]. The low albumin levels observed in this study suggest metabolic dysregulation associated with DM, such as impaired hepatic protein synthesis or increased albumin turnover caused by chronic inflammation or oxidative stress. Moderately increased urinary albumin excretion may serve as an early marker of diabetic nephropathy [23], which aligns with the increase in BUN and creatinine levels observed in the present study. This finding is consistent with that of a previous study [24] and underscores the complex nature of diabetic complications, which affect both renal and hepatic systems.

In the present study, patients with DM exhibited higher levels of alkaline phosphatase, triglycerides, and glycohemoglobin. High alkaline phosphatase levels, a indicative of hepatobiliary or bone disorders [25], are associated with inflammatory markers in adults [26]. Moreover, increased levels of triglycerides and glycohemoglobin indicate metabolic disturbances, such as dyslipidemia and poor glycemic control, which are commonly observed in patients with diabetes and contribute to increased cardiovascular risks [27].

Our findings have crucial implications for both clinical practice and public health initiatives. Given the increased mortality risk associated with DM in individuals with sarcopenic obesity, targeted interventions focusing on diabetes management, physical activity promotion, and nutritional support can mitigate adverse outcomes in these individuals. Strategies for preserving muscle mass and reducing fat mass, particularly among older adults with diabetes, can improve long-term health outcomes and quality of life.

The limitations of this study include its retrospective design and reliance on mortality records for cause-specific analyses, which may introduce biases and limit the ability to draw causal conclusions. Prospective cohort studies with detailed

clinical assessments and imaging modalities are required to explore the mechanistic links between DM, sarcopenic obesity, and specific cerebrovascular outcomes. Furthermore, investigating the role of emerging biomarkers and genetic predispositions may offer further insights into individual susceptibility to diabetes-related vascular complications.

## Conclusion

In this study, we analyzed NHANES data linked with mortality records and discovered that DM significantly increases the risk of mortality in patients with HA-LM, primarily due to cardiovascular disease. Although an increased BMI is associated with altered metabolic markers, the specific effect of DM on body composition indices, such as the ASMI and FMI, in sarcopenic obesity remains unclear. Targeted interventions focusing on diabetes management and muscle mass preservation can mitigate adverse outcomes in this population. Additional prospective studies are required to optimize preventive strategies.

## Acknowledgments

This manuscript was edited by Wallace Academic Editing.

## Author contributions

**Conceptualization:** Ting-Ju Kuo, Shih-Wei Huang, Hui-Wen Lin.

**Formal analysis:** Shih-Wei Huang.

**Methodology:** Ting-Ju Kuo, Shih-Wei Huang, Hui-Wen Lin.

**Supervision:** Hui-Wen Lin.

**Writing – original draft:** Ting-Ju Kuo.

**Writing – review & editing:** Shih-Wei Huang, Hui-Wen Lin.

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
