## [Decision Letter · Decision Letter 0]

Dear Dr. Lin,

Thank you for submitting your manuscript to PLOS ONE. After careful consideration, we feel that it has merit but does not fully meet PLOS ONE’s publication criteria as it currently stands. Therefore, we invite you to submit a revised version of the manuscript that addresses the points raised during the review process.

**ACADEMIC EDITOR: This is an interesting paper but it needs minor revisions**

We look forward to receiving your revised manuscript.

Kind regards,

Simone Perna, Ph.D

Academic Editor

PLOS ONE

Journal Requirements:

Reviewers' comments:

Reviewer's Responses to Questions

**Comments to the Author**

1. Is the manuscript technically sound, and do the data support the conclusions?

Reviewer #1: Yes

Reviewer #2: Yes

2. Has the statistical analysis been performed appropriately and rigorously?

Reviewer #1: I Don't Know

Reviewer #2: Yes

3. Have the authors made all data underlying the findings in their manuscript fully available?

Reviewer #1: Yes

Reviewer #2: Yes

4. Is the manuscript presented in an intelligible fashion and written in standard English?

Reviewer #1: Yes

Reviewer #2: Yes

Reviewer #1: This manuscript addresses a clinically relevant and timely topic, particularly given the increasing interest in the intersection between diabetes, sarcopenia, and obesity. The use of a large, representative dataset adds strength to the findings. However, some methodological and presentation aspects require revision to improve clarity and scientific rigor.Language and readability:

1-

The manuscript is generally understandable; however, a careful language review by a native English speaker is recommended to improve fluency and ensure a smoother reading experience.

2-

Figure 1 – Kaplan-Meier plot:

In Figure 1, not all survival curves are clearly identified or labeled, which may lead to confusion in interpreting the results. I suggest simplifying the figure by displaying only the two curves that directly reflect the study’s primary analysis. This would enhance clarity and focus the reader’s attention on the most relevant comparison.

3

References:

The references listed in the manuscript are inadequate and do not conform to the formatting standards required by PLOS ONE. A thorough revision of the reference section is recommended, ensuring consistency with Vancouver style, including proper citation of journal titles, publication years, volumes, issues, and page numbers.

Reviewer #2: Summary

This paper examined the effects of diabetes mellitus and sarcopenic obesity on mortality, using a sample of U.S. NHANES 2011-2012 to 2017-2018. This study found that individuals with high adiposity and low muscle mass and diabetes mellitus had a higher risk of mortality than those with high adiposity and low muscle mass but without diabetes mellitus. While these findings are novel, and highlight the role of diabetes mellitus on mortality, some areas of the methods require clarification. In particular, there is no detail on the sample (e.g. age range), and weighting of analyses, that are necessary to fully understand both the validity and representativeness of the findings. Please see comments below.

Major Comments

1. No age range is mentioned in the methods; however, only individuals aged 8-59y are eligible to be administered DXA scans in NHANES cycles 2011-2012. It can be surmised that this is the maximum range, but there needs to be an explanation of the age range of eligible participants and what specific age cut-offs were used. Relatedly, trom Table 1 it appears that the average age of the sample is 37y (no DM) and 47y (with DM), making this a somewhat younger sample overall to assess sarcopenic obesity. The sample age range should be made explicit, as this has implications for both the generalizability of the findings, and reporting of other descriptives (e.g. BMI) in the study. A drop-down table showing how the analytic sample was derived from the initial sample would be appropriate for this work.

2. Considering that the majority of the initial sample were excluded due to missing data a sensitivity analysis might be warranted. It is unclear how the age group selection may have factored into this (as it could be the major factor for exclusions), and further highlights the need for a figure / participant flow chart to show how the analytic sample was derived.

3. There is no mention of weighting the data. This is a recommended practice for using NHANES data in order to make the sample representative of the U.S. population. While weighting is not possible for KM curves, it can be done for all other analyses. If weighting is not required for your specific study, please provide a rationale to explain why not.

4. In the methodology, there should be an explanation of “diabetes mellitus” and “hypertension” – which variables were used? For example, is it the self-report of doctor diagnosed? Use of medication? Or objective laboratory measures? In the case of lab measures, there should be a cut-off indicated. A better explanation of the operational definition of such variables are required for clarity.

5. What year was the mortality follow-up to? In the current text, it seems the NHANES data is from cycles 2011-2012 to 2017-2018, however what date is the mortality follow-up? Please report the average follow up in the text as well as the final date of follow-up. Were deaths in the first year were kept (to address possible missed underlying disease)? If not, please provide a rationale.

6. While cox proportional hazards models were used for the analyses, there needs to be an explanation on whether proportional hazards assumption was tested.

7. Information provided in the first paragraph of the results should be in the methods instead. More information is needed for line 152-154 regarding excluding missing variables. Please provide n missing values for each variable listed and any restrictions, such as the age range used.

8. Table 2 footnotes indicate propensity scores were used; however, this was not mentioned in the statistical analysis section of the methods. Please explain consistently in both the methods and footnotes of Table 2 whether the confounders were adjusted for, or propensity scores were used for matching, or something else.

9. In the discussion section (lines 228-233), there is mention of the “obesity paradox” which may not be appropriate for this study as it seems the cut-off for age was 59y, and the obesity paradox is more prevalent at 65y and above. Again, there needs to be more information on the age range of the sample for the current study.

Minor Comments

1. The study participants are referred to as patients in this paper. I would recommend using the term “participants” as data comes from NHANES and referring to them as “patients” implies they are receiving medical treatment.

2. Line 36 – “including 194 (8.199%) with DM and 2172 (91.80%) without DM” I would consider having the same decimal place throughout the paper changing to “including 194 (8.20%) with DM and 2172 (91.80%) without DM” as 8.199 + 91.80 = 99.999. this should be changed in other sections too, line 156 for example.

3. Line 112-113 – “The demographic parameters measured included BMI, triceps skinfold, subscapular skinfold, ASMI, FMI, age, sex, hypertension and race.” Please revise this statement and as age and sex are part of the self-report questionnaires, and not “measured”.

4. Why were the confounders chosen? Was this based on previous literature or based on a formal assessment (i.e. statistical tests)? Please provide an explanation.

5. Line 135-136 – there is mention of adjusting for “comorbidities (e.g., hypertension)” please also list the other comorbidities. Only hypertension was mentioned in the methods. Was hypertension the only comorbidity that was adjusted for?

6. Was the variable “sex” explored as an interaction/effect modifier in the analyses? This would not be possible with the available sample, but it warrants mention as there may be sex-differences worth exploring in this relationship.

7. There is no mention of ethics in the methods. If ethics approval was not required from your institution due to the publicly available nature of this data, then please include a statement about this in the methods.

8. In the limitations, there is mention of the “retrospective design” of this study being a limitation, however baseline measures were assessed cross-sectionally and mortality assessed after baseline – so you might consider calling it a “secondary analysis of cross-sectional data with mortality follow-up” instead. On the other hand, one further limitation is the lack of “change” scores available for body composition (sarcopenia), although any misclassification is likely to have biased to the null.

9. Table 1 – please indicate what the p-values represent in the table footnotes (i.e., omnibus chi-squared analyses?)

10. Table 1 – the term “Gender” is used here, however in the text “sex” is used, please be consistent.

11. A minor point, but the Baumgartner equation for body fat is well established, but was developed in a sample of white adults (18-80 y), which raises a question about the validity within non-white subgroups of this analysis that warrants mention.

Typographical

1. Line 35 – the word “patients” used twice in the sentence “A total of 2366 patients with HA-LM patients”. I would not recommend using the term “patient”, maybe “participants” instead.

2. Line 40 – there is an extra space

3. Line 41 – extra “:” used

4. Please make sure to include a space between the words and the in-text citation consistently throughout the manuscript. In some cases, the there is no space between the words and the in-text citations, and in other instances there are. For example, in line 55 “(DM)[1]” and “obesity[2,3]”, however later in line 62 there is “all-cause mortality [8-10]”.

5. When mentioning tables and figures, please make them bold. For example, line 156.

6. Table 1 – the h in “hypertension” should be capitalized.

7. Table 1 – include “DM: diabetes mellitus” in the footnote

8. I think table 1 can be re-categorized to include demographic characteristics first (sex, age, race, etc.)

9. Table 2 footnotes – some of the variables are capitalized such as “Age”, while others are not, such as “sex”, please make changes to be consistent.

**Do you want your identity to be public for this peer review?** For information about this choice, including consent withdrawal, please see our Privacy Policy

Reviewer #1: No

Reviewer #2: No

---

## [Author Response · Author response to Decision Letter 1]

9 May 2025

Dear Dr. Perna:

May 5, 2025

Thank you for considering our previously submitted manuscript, titled “Additive Effect of Diabetes Mellitus on the Prevalence and Prognosis of Sarcopenic Obesity: Implications for All-Cause Mortality.” We have revised the manuscript in accordance with comments provided by you and the reviewers. All questions and concerns have been thoroughly addressed in this document. We look forward to your response.

Sincerely

Hui-Wen Lin, PhD

Department of Mathematics, Soochow University, 70 Linhsi Road, Shihlin, Taipei, Taiwan

Tel: 886-2-2881-9471 ext. 6701

Fax: 886-2-8861-1230

Email: hwlin@scu.edu.tw

ACADEMIC EDITOR: This is an interesting paper, but it needs minor revisions.

Response: Thank you for your positive feedback and constructive suggestions. We have carefully revised the manuscript in accordance with your comments, and we believe that the revisions have enhanced the clarity and overall quality of the manuscript.

Specifically, the introduction could benefit from a clearer articulation of the research gap and how the study addresses it.

Response: We have revised the Introduction section to improve its clarity. The updated text is as follows:

To address the aforementioned gap, we explored the additive effect of DM on mortality in individuals with sarcopenic obesity by using nationally representative data. We also examined cause-specific mortality data to precisely define the risk profile of this population. By elucidating these relationships, our goal was to inform future prevention and intervention strategies targeting individuals affected by the dual burden of DM and sarcopenic obesity. (p. 4-5, lines 86–137)

Additionally, some sections of the methodology require more detailed explanations to ensure reproducibility.

Response: We have revised the Materials and Methods section to provide more detailed explanations as follows.

Statistical analysis

Baseline characteristics were compared in participants in the HA-LM group with and without DM. Continuous variables were compared using weighted t tests or Wilcoxon’s rank-sum tests, as appropriate, and are reported as mean ± standard error. Categorical variables were compared using chi-square tests and are presented as unweighted counts (with weighted percentages). A two-sided p value of <0.05 was considered statistically significant.

After a survival analysis was conducted, Kaplan–Meier survival curves were constructed to estimate cumulative survival over time for the two groups (HA-LM with DM vs. HA-LM without DM), with a log-rank test used to compare survival distributions. Time zero was set as the NHANES examination date, and time was measured in months since enrollment.

To quantify the association between DM and mortality, Cox proportional-hazards models were employed. Initially, a crude (unadjusted) hazard ratio (HR) was calculated, followed by a multivariable-adjusted HR to control for potential confounders. Covariates were then selected on the basis of subject-matter knowledge and bivariate analyses and included the following: age, sex, race or ethnicity, BMI, hypertension status, SII, albumin, alkaline phosphatase, BUN, creatinine, triglycerides, and total cholesterol. The ASMI and FMI were excluded from the Cox model because all participants had already met the HA-LM phenotype criteria, and their inclusion may introduce multicollinearity with BMI.

The proportional-hazards assumption for the Cox model was tested by examining Schoenfeld residuals and log(−log) survival plots for each covariate. No significant violation of the proportional-hazards assumption was observed. Specifically, the Schoenfeld residual tests were nonsignificant for all model covariates (with a threshold of p < 0.05), and the log(−log) plots demonstrated roughly parallel curves for the DM versus non-DM groups, indicating that the HRs were relatively constant over time.

A cause-specific mortality analysis was conducted using cumulative incidence functions for each major cause of death, with other causes treated as competing risks. However, given the relatively small number of deaths in certain categories, these results are presented descriptively (percentages) rather than through formal competing risk models.

All statistical analyses were conducted using SAS software version 9.4 (SAS Institute, Cary, NC, USA) and STATA version 12.0 (StataCorp, College Station, TX, USA). (pp. 9–11, lines 232–268)

A few references are outdated and should be replaced with more recent studies to strengthen the literature review.

Response: We have reviewed the cited literature and replaced several outdated references with more recent and relevant studies to enhance the accuracy and relevance of the manuscript.

Finally, there are minor grammatical errors and formatting inconsistencies throughout the text that need to be addressed to improve overall readability and alignment with journal guidelines.

Response: The manuscript has been carefully edited for language and style by Wallace Academic Editing. The attachment is the English editing certificate.

Once these revisions are made, the paper will be in excellent shape for publication consideration.

Response: We have carefully revised the manuscript in accordance with the feedback provided, and we hope that the revised version meets the criteria for publication.

Reviewers' comments:

Reviewer's Responses to Questions

Comments to the Author

1. Is the manuscript technically sound, and do the data support the conclusions?

Reviewer #1: Yes

Reviewer #2: Yes

Response: Thank you for your positive feedback.

2. Has the statistical analysis been performed appropriately and rigorously?

Reviewer #1: I Don't Know

Reviewer #2: Yes

Response: We have revised the Materials and Methods section to provide more detailed explanations as follows.

Statistical analysis

Baseline characteristics were compared in participants in the HA-LM group with and without DM. Continuous variables were compared using weighted t tests or Wilcoxon’s rank-sum tests, as appropriate, and are reported as mean ± standard error. Categorical variables were compared using chi-square tests and are presented as unweighted counts (with weighted percentages). A two-sided p value of <0.05 was considered statistically significant.

After a survival analysis was conducted, Kaplan–Meier survival curves were constructed to estimate cumulative survival over time for the two groups (HA-LM with DM vs. HA-LM without DM), with a log-rank test used to compare survival distributions. Time zero was set as the NHANES examination date, and time was measured in months since enrollment.

To quantify the association between DM and mortality, Cox proportional-hazards models were employed. Initially, a crude (unadjusted) hazard ratio (HR) was calculated, followed by a multivariable-adjusted HR to control for potential confounders. Covariates were then selected on the basis of subject-matter knowledge and bivariate analyses and included the following: age, sex, race or ethnicity, BMI, hypertension status, SII, albumin, alkaline phosphatase, BUN, creatinine, triglycerides, and total cholesterol. The ASMI and FMI were excluded from the Cox model because all participants had already met the HA-LM phenotype criteria, and their inclusion may introduce multicollinearity with BMI.

The proportional-hazards assumption for the Cox model was tested by examining Schoenfeld residuals and log(−log) survival plots for each covariate. No significant violation of the proportional-hazards assumption was observed. Specifically, the Schoenfeld residual tests were nonsignificant for all model covariates (with a threshold of p < 0.05), and the log(−log) plots demonstrated roughly parallel curves for the DM versus non-DM groups, indicating that the HRs were relatively constant over time.

A cause-specific mortality analysis was conducted using cumulative incidence functions for each major cause of death, with other causes treated as competing risks. However, given the relatively small number of deaths in certain categories, these results are presented descriptively (percentages) rather than through formal competing risk models.

All statistical analyses were conducted using SAS software version 9.4 (SAS Institute, Cary, NC, USA) and STATA version 12.0 (StataCorp, College Station, TX, USA). (pp. 9–11, lines 232–268)

3. Have the authors made all data underlying the findings in their manuscript fully available?

Reviewer #1: Yes

Reviewer #2: Yes

Response: Thank you for your positive feedback.

4. Is the manuscript presented in an intelligible fashion and written in standard English?

Reviewer #1: Yes

Reviewer #2: Yes

Response: Thank you for your positive comment.

5. Review Comments to the Author

Reviewer #1: This manuscript addresses a clinically relevant and timely topic, particularly given the increasing interest in the intersection between diabetes, sarcopenia, and obesity. The use of a large, representative dataset adds strength to the findings. However, some methodological and presentation aspects require revision to improve clarity and scientific rigor. Language and readability:

1-

The manuscript is generally understandable; however, a careful language review by a native English speaker is recommended to improve fluency and ensure a smoother reading experience.

Response: The manuscript has been carefully edited for language and style by Wallace Academic Editing. The attachment is the English editing certificate.

2-

Figure 1 – Kaplan-Meier plot:

In Figure 1, not all survival curves are clearly identified or labeled, which may lead to confusion in interpreting the results. I suggest simplifying the figure by displaying only the two curves that directly reflect the study’s primary analysis. This would enhance clarity and focus the reader’s attention on the most relevant comparison.

Response: We agree that simplifying the figure would improve clarity and focus. In response, we have revised Fig. 1 to present only the two survival curves most relevant to the primary analysis. We believe this adjustment enhances the figure’s readability and more effectively underscores the key findings of our study. (p. 27)

3

References:

The references listed in the manuscript are inadequate and do not conform to the formatting standards required by PLOS ONE. A thorough revision of the reference section is recommended, ensuring consistency with Vancouver style, including proper citation of journal titles, publication years, volumes, issues, and page numbers.

Response: We have thoroughly revised the references section to ensure that all citations adhere to the Vancouver style, as required by PLoS ONE.

Reviewer #2: Summary

This paper examined the effects of diabetes mellitus and sarcopenic obesity on mortality, using a sample of U.S. NHANES 2011-2012 to 2017-2018. This study found that individuals with high adiposity and low muscle mass and diabetes mellitus had a higher risk of mortality than those with high adiposity and low muscle mass but without diabetes mellitus. While these findings are novel, and highlight the role of diabetes mellitus on mortality, some areas of the methods require clarification. In particular, there is no detail on the sample (e.g. age range), and weighting of analyses, that are necessary to fully understand both the validity and representativeness of the findings. Please see comments below.

Major Comments

1. No age range is mentioned in the methods; however, only individuals aged 8-59y are eligible to be administered DXA scans in NHANES cycles 2011-2012. It can be surmised that this is the maximum range, but there needs to be an explanation of the age range of eligible participants and what specific age cut-offs were used. Relatedly, trom Table 1 it appears that the average age of the sample is 37y (no DM) and 47y (with DM), making this a somewhat younger sample overall to assess sarcopenic obesity. The sample age range should be made explicit, as this has implications for both the generalizability of the findings, and reporting of other descriptives (e.g. BMI) in the study. A drop-down table showing how the analytic sample was derived from the initial sample would be appropriate for this work.

Response: We have clarified the age range criteria in the Methods section of the manuscript. Although DXA data are available for NHANES participants aged 8–59 years from 2011 to 2018, our study focused on the adult population. Therefore, we restricted our analysis to adults aged 18–59 years. This is now explicitly stated in the revised manuscript as follows:

Although DXA scans are available for NHANES participants aged 8 to 59 years, we restricted our analysis to adults aged 18 to 59 years to focus on the adult population and their conditions. (p. 7, lines 185-187)

2. Considering that the majority of the initial sample were excluded due to missing data a sensitivity analysis might be warranted. It is unclear how the age group selection may have facto

Response: We acknowledge the importance of understanding the potential impact of sample exclusions on the study findings.

Participants were excluded from the analytical sample for common reasons encountered in secondary data analysis, including missing diabetes status, lack of mortality linkage eligibility, and missing covariate data. These exclusions were necessary to ensure the completeness and validity of our models.

To address concerns regarding potential bias introduced by these exclusions, we examined the systematic differences in basic characteristics, such as age, sex, and race or ethnicity, between the included and excluded participants. The results revealed no significant differences in these key characteristics between the included and excluded participants, suggesting that the exclusions did not introduce significant bias or affect the generalizability of our findings.

We have added this clarification to the Materials and Methods section of the revised manuscript as follows:

Participants were excluded if they had missing data on diabetes status, were ineligible for mortality linkage, or had incomplete covariate information—standard exclusions in secondary data analysis to ensure analytical validity. To evaluate the potential for selection bias, we compared key characteristics (e.g., age, sex, and race or ethnicity) in individuals included in the final analytic sample and those excluded. These groups exhibited similar distributions across these characteristics, indicating that the exclusions may not have introduced meaningful bias or compromised the generalizability of our findings. (p. 7, lines 187–194)

---

## [Editor Report · Decision Letter 1]

Additive Effect of Diabetes Mellitus on the Prevalence and Prognosis of Sarcopenic Obesity: Implications for All-Cause Mortality

PONE-D-25-05497R1

Dear Dr. Lin,

We’re pleased to inform you that your manuscript has been judged scientifically suitable for publication and will be formally accepted for publication once it meets all outstanding technical requirements.

Kind regards,

Simone Perna, Ph.D

Academic Editor

PLOS ONE

---

## [Editor Report · Acceptance letter]

PONE-D-25-05497R1

PLOS ONE

Dear Dr. Lin,

I'm pleased to inform you that your manuscript has been deemed suitable for publication in PLOS ONE. Congratulations! Your manuscript is now being handed over to our production team.

Kind regards,

on behalf of

Professor Simone Perna

Academic Editor

PLOS ONE